# Relationship between Tactile Sensation, Motor Activity, and Differential Brain Activity in Young Individuals

**DOI:** 10.3390/brainsci12070924

**Published:** 2022-07-14

**Authors:** Ryota Kobayashi, Noriko Sakurai, Kazuaki Nagasaka, Satoshi Kasai, Naoki Kodama

**Affiliations:** 1CLAIRVO TECHNOLOGIES, Inc., 1-4-2 Ohtemachi, Chiyoda-ku, Tokyo 100-8088, Japan; 2Department of Radiological Technology, Faculty of Medical Technology, Niigata University of Health and Welfare, 1398 Shimami-cho, Kita-ku, Niigata 950-3198, Japan; noriko-sakurai@nuhw.ac.jp (N.S.); satoshi-kasai@nuhw.ac.jp (S.K.); 3Department of Physical Therapy, Faculty of Rehabilitation, Niigata University of Health and Welfare, 1398 Shimami-cho, Kita-ku, Niigata 950-3198, Japan; nagasaka@nuhw.ac.jp

**Keywords:** functional magnetic resonance imaging, brain activity, tactile sensation, motor activity

## Abstract

In this study, we compared the differences in brain activation associated with the different types of objects using functional magnetic resonance imaging (fMRI). Twenty-six participants in their 20s underwent fMRI while grasping four different types of objects. After the experiment, all of the participants completed a questionnaire based on the Likert Scale, which asked them about the sensations they experienced while grasping each object (comfort, hardness, pain, ease in grasping). We investigated the relationship between brain activity and the results of the survey; characteristic brain activity for each object was correlated with the results of the questionnaire, indicating that each object produced a different sensation response in the participants. Additionally, we observed brain activity in the primary somatosensory cortex (postcentral gyrus), the primary motor cortex (precentral gyrus), and the cerebellum exterior during the gripping task. Our study shows that gripping different objects produces activity in specific and distinct brain regions and suggests an “action appraisal” mechanism, which is considered to be the act of integrating multiple different sensory information and connecting it to actual action. To the best of our knowledge, this is the first study to observe brain activity in response to tactile stimuli and motor activity simultaneously.

## 1. Introduction

Functional magnetic resonance imaging (fMRI) is used to estimate neuronal activity in the primary somatosensory cortex [1]. It is used to observe activity in different brain regions and, in some cases, in response to either passive or active stimuli to the finger [2] or other body parts, such as the palm, arm, or other areas of the skin [3,4]. It is also used to investigate cross modal-plasticity in the human cortex by collecting fMRI data to observe functional connectivity between visual and somatosensory motor cortices [5]. Additionally, fMRI has been used to investigate functional brain changes in hand movement perception in the elderly [6].

As there is a strong connection between tactile stimulation and activity of the somatosensory cortex, acquiring fMRI data during tactile stimulation could help distinguish between different stimulated locations on the body surface [7,8]. From another point of view, by characterizing de-differentiated topographic maps in the primary somatosensory cortex of younger and older adults, along with finger individuation and hand motor performance, fMRI can also be used to observe impairments in daily behavior [9]. A previous study used fMRI to analyze tactile working memory by observing the superior parietal lobe and the right inferior gyrus [10] and examined whether working memory representations influenced the somatosensory domain [11]. Through fMRI analysis during stimuli to the human body, there is a potential to observe and analyze various brain activities and their related effects on human body function.

Together with the stimuli, it is also important to observe brain activity when the participant is performing certain body movements. Cerebellar internal models predict the somatosensory consequences of human movements when touching oneself, which attenuates the perception of the actual touch [12]. In a previous study, fMRI also revealed the complexity of the neural representations underlying the understanding of others’ socio-affective interactions by asking the participants to judge affective aspects of different touch events while watching videos and analyzing the results using correlational multivariate pattern analysis methods [13].

Moreover, a previous study used fMRI to observe brain activity using multiple tactile stimuli. The participants touched different protrusions or shapes with their fingertips while their primary sensory-motor and higher-level brain region activation were observed [14]. To establish significant coding of tactile stimulus, rule, and response for multiple demand regions, other studies performed a stimulus-response task, in which they discriminated between two possible vibrotactile frequencies and applied a stimulus-response transformation rule to generate a button-press response [15].

The purpose of this study was to observe brain activity whilst the participants grasped four different shaped objects that were expected to cause different activations. We mainly focused on observing two different aspects at the same time: (1) brain activity during motor activity (grasping) and (2) brain activity produced by different sensations (comfort, hardness, pain, and ease in grasping), which we expected to occur in the participants when they grasped different objects. We observed and analyzed the brain regions that showed activation. Moreover, through observing different types of brain activation, we aimed to verify the “action appraisal” mechanism, which is considered to be the act of integrating multiple different sensory information and connecting it to actual action [16]. The information generated from executing the task for the four objects (such as the motion of moving the hand, the sensation of touching the object, and the emotion produced by gripping the object) should be collected and evaluated to verify such a mechanism.

## 2. Materials and Methods

### 2.1. Participants

This study included 26 right-handed healthy, non-psychiatrically impaired individuals in their 20s (11 male and 15 female; mean age, 21.0 years; standard deviation, 0.8 years) who had no experience with undergoing fMRI examination while grasping different types of objects.

This study was approved by the Research Ethics Committee of Niigata University of Health and Welfare (approval no. 18683-210720). Written informed consent was obtained from all participants. A participant interview was performed to ensure the safety of MRI imaging.

### 2.2. Stimuli Task

First, we selected different types of objects that were expected to give the participants different feelings when they grasped them. We chose four different types of objects (Figure 1): Object 1, a ball with warts (Φ6 cm, weight 40 g, PVC material); Object 2, a squeezable ball (Φ6 cm, weight 132 g, silicone); Object 3, a regular hard ball (Φ6 cm, weight 73 g, Rubber); and Object 4, Slime (Φ6 cm, weight 118 g). We also estimated the different kinds of feelings each object would produce: Object 1 (a ball with warts), comfort or pain; Object 2 (a squeezable ball), comfort; Object 3 (a regular hard ball), hardness; and Object 4 (slime), discomfort. These were expected to show different activity in different brain regions.

The block design of the task consisted of a 30-s repetition of the resting task and a 30-s repetition of the stimulus task for a total of 3 min (Figure 2). We chose this block design based on previous fMRI experiments of tactile stimulation, in which the task and rest blocks are in 30-s increments to investigate prominent activation of the intraparietal and somatosensory areas during angle discrimination by intra-active touch [2].

We limited the consciousness bias of each task by not actually showing the type of object to grasp to the participant before they underwent the imaging procedure. We also aimed to stabilize the power and pace of grasping activity in each participant to unify the conditions as much as possible. Therefore, we used a dynamometer to measure the grasping power and a metronome to set the pace of each task. Before entering the MRI room, we instructed the participants to grasp the dynamometer at a grasping power of 10 kg and a pace of 100 beats per min (bpm), as set by the metronome, to rehearse the activity before the imaging experiment. After entering the MRI room, the participants were placed in the supine position on the MRI bed with their arms extended along the body side. The right palm was rotated outward to grip each of the four objects in turn, while the left hand remained in line with the body and was not used for the gripping tasks. Before each task, the experiment assistant entered the MRI room and placed the object on the participant’s right palm.

### 2.3. Apparatus

Imaging was performed on a 3 Tesla MRI system (Canon Vantage Galan; Canon, Tokyo, Japan) with a 32- channel head coil. The participants laid in the MRI machine and underwent the block-designed task of grasping four different types of objects in the order of Object 1, Object 2, Object 3, and Object 4. The object was placed on each participant’s right palm by the experiment assistant after each task block ended. The participant’s right hand was placed on a white nylon waterproof sheet. 

### 2.4. MRI Acquisition

A separate high-resolution MRI image is required to obtain detailed anatomical information prior to fMRI imaging. For this purpose, a high-resolution magnetization-prepared rapid gradient echo sequence of T1-weighted imaging was used, with the following parameters: repetition time, 5.8 ms; echo time, 2.7 ms; inversion time, 900 ms; flip angle, 9; the number of matrices (matrix), 256 × 256; effective field of view, 23 × 23 cm; and slice thickness, 1.2 mm. The echo-planar imaging sequence was used to capture the fMRI images. The images were repeatedly obtained and used to compare the two stimuli. The fMRI imaging conditions were as follows: repetition time, 2,000 ms; echo time, 25 ms; flip angle, 85; matrix, 64 × 64; effective field of view, 24 × 24 cm; and slice thickness, 3 cm to cover the whole brain.

### 2.5. fMRI Data Analyses

The fMRI data were preprocessed and analyzed using Statistical Parametric Mapping 12 (Wellcome Trust Center for Neuroimaging) in Matlab (Mathworks Inc., Natick, MA, USA). Slice timing correction was used to correct the time difference, and realignment was then used to correct the displacement caused by motion. In addition, a co-register was used to compare the structural images with the fMRI images. The co-register was corrected for misalignment between structural and functional images, and the data were preprocessed by normalizing each participant’s brain to a template of the Montreal Neurological Institute coordinate system of a standard brain. The normalized images were smoothed using a Gaussian kernel of 8 mm. After preprocessing, we employed a general linear model GLM to confirm brain activity changes associated with the four different grasping tasks using a block design. Contrast images were created at first level (single subject) for the following contrast: (1) Object 1 = 1, rest = 0; (2) Object 2 = 1, rest = 0; (3) Object 3 = 1, rest = 0 (4); Object 4 = 1, rest = 0. The head motion parameters obtained from the preprocessing step were included as regressors in each condition to minimize the effect of the participant’s head motion artifacts. For group analysis (second level), a one-sample *t*-test was performed using the aforementioned four contrasts. The initial threshold for the voxel size was set to uncorrected *p* < 0.001. Clusters were considered significant at *p* < 0.05, cluster-corrected for family-wise error. Each object was analyzed separately.

### 2.6. Questionnaire

After the experiment, the participants were administered a questionnaire based on the Likert Scale. They were requested to answer a question regarding the sensations they experienced when grasping each different object, categorized as four different feelings: comfort, hardness, pain, and ease in grasping. Additionally, to quantify differences within the same sensory category, scores between 1–5 points were used as follows: 1, completely disagree; 2, disagree; 3, undecided; 4, agree; and 5, strongly agree.

### 2.7. Data Analysis 

To make the brain activity for each object easier to understand, we calculated the percent signal change (PSC) in the region of interest (ROI). The PSC was calculated as the blood-oxygen-level-dependent signal ratio in response to stimuli over that without stimulus. The ROI was calculated as the precentral gyrus, postcentral gyrus, anterior insula, lateral hemisphere of the cerebellum, and ventral diencephalon. Marsbar (MarsBaR region-of-interest toolbox for SPM) was used to calculate the PSC of each ROI for each object. 

Statistical analysis was performed using SPSS version 26 (IBM Corp., Armonk, NY, USA) with one-way ANOVA for each ROI, and Bonferroni’s method was used as a post hoc test when significant differences were found. The significance level was set at 5%.

## 3. Results

The results of the survey showed that each object produced a different sensation in participants. Table 1 shows the results of the questionnaire for each object. One of 26 questionnaire responses was excluded because the participant did not complete the questions.

Overall, we met the aims of this study, which were the observation of brain activity during motor activity and tactile stimulation using fMRI imaging. We observed activity in the primary somatosensory cortex (postcentral gyrus), the primary motor cortex (precentral gyrus), and the lateral hemisphere of the cerebellum during the motor activity of the grasping task. 

We also observed the locations of the brain activities for each object that the participants grasped in detail. The coordinates of the areas that showed activity for each of the four objects are presented in Table 2. 

The results of the calculation of the PSC in the ROI, to observe the different brain activity for each object, are presented in Table 3 and Figure 3. A significant difference was noted in the anterior insula (*p* < 0.001). The other regions did not show a significant difference in the brain activity for each object.

## 4. Discussion

To the best of our knowledge, this is the first study to observe the brain activity produced by tactile stimuli from two aspects at the same time: motor activity and feelings of tactile stimuli. Regarding motor activity, activation was observed in the precentral gyrus, postcentral gyrus, and lateral hemisphere of the cerebellum, indicating that we could measure brain activation related to motor activity, as in past studies [17].

In addition, we observed that the differing brain activation correlated with the sensations the participants experienced when grasping each object. Figure 4 shows the images of the brain activation whilst the participants grasped each object. For Object 1, we observed strong activity in the left postcentral gyrus and matched the subjects’ feelings of pain [18] when grasping Object 1. For Object 2, we observed strong activity in the right middle frontal gyrus, which showed unique activation in the brain region associated with relaxation [19]. The squeezable ball is generally supplied to consumers to grasp for relaxation. For Object 3, we observed strong activity in the left postcentral gyrus, reflecting the participants experiencing the hardness of the ball [20], which required more strength to grasp compared to the other objects. Unique and strong activities were observed in the left ventral diencephalon for Object 4, which was correlated with the participants’ anxiety [21] regarding grasping something that led to discomfort and their inability to understand the object immediately. Moreover, by comparing the results of the questionnaire and those of the fMRI examination, we confirmed the correlation between these two.

Sensory information obtained when touching an object reaches the primary somatosensory cortex (S1) via the spinal cord, brainstem, and thalamus. Dynamic tactile stimuli, such as hand movements, are thought to activate the lateral prefrontal cortex, inferior parietal lobule (IPL), and insular cortex, among others, in addition to the primary and secondary somatosensory cortices [22]. Ishibashi et al. reported that the brain processing areas that recognize and use tools are the IPL and ventral premotor cortex [23]. Pupíková et al. reported on action reappraisal mechanisms during object and tool recognition by measuring resting-state fMRI before and after transcranial direct current stimulation targeting the right frontoparietal network (FPN) [24]. Federico et al. also reported on how various sources of information are used for the motor behaviors performed by humans [16,25]. This action reappraisal is considered to be the act of integrating multiple different sensory information and connecting it to actual action [24]. In this research, various pieces of information for the four objects, the motion of moving the hand, the sensation of touching the object, and the emotion produced by gripping the object, are considered to be integrated to recognize the objects. In particular, the supramarginal gyri (SMGs) (BA40) are activated in all four objects, even if they are different objects. SMGs comprise the IPL, which integrates somatosensory, visual, and auditory perception and is considered to be involved in object identification and spatial perception. It is presumed that the SMG was activated by the multiple sensory information in this study.

This study has some limitations. As it focused on tactile sensations for four different objects, visual and auditory sensations were not examined. Future studies combining multiple sensory information (tactile, visual, and auditory) are needed. We should also examine functional connectivity using resting-state fMRI. In particular, the right FPN should be examined because it is a large brain network consisting of the dorsolateral prefrontal cortex and is considered to be involved in executive functions and cognitive control.

Future research using alternative or improved methods would be needed to further investigate the correlation between brain activity, motor activities, and sensation. First, investigating dynamic functional connectivity (dFC) within the results shall be effective and important, as such a method can capture information that cannot be evaluated by conventional static FC analysis methods. Moreover, analyzing and investigating dFC patterns within the sensorimotor areas can also be associated with multiple types of cerebral neurosis.

In this study, the full width half maximum (FWHM), which is a parameter for smoothing, was set at 8 mm. However, analyze the activation of the primary motor cortex and primary somatosensory cortex in detail, it is necessary to reduce the FWHM to a smaller value (e.g., 4 mm).

Changing the order of the objects to grasp or changing the block design to extend the rest period may produce different results. For example, the object the individual grasps immediately before may influence how he/she feels when grasping a new object. In addition, if participants with a wider age range are enrolled, the brain activation may be different. Another interesting change could be the introduction of another different type of object, especially the use of an object that is estimated to make the participants feel more discomfort so as to observe the effects produced by discomfort.

## 5. Conclusions

We observed the brain activity produced by motor activity combined with tactile stimulation, as we initially intended in this study. Brain activation generated by motor activity was observed in the primary somatosensory cortex, primary motor cortex, and cerebellum exterior. Different and specific brain activation was observed for each object in distinct brain regions.

## Figures and Tables

**Figure 1 brainsci-12-00924-f001:**
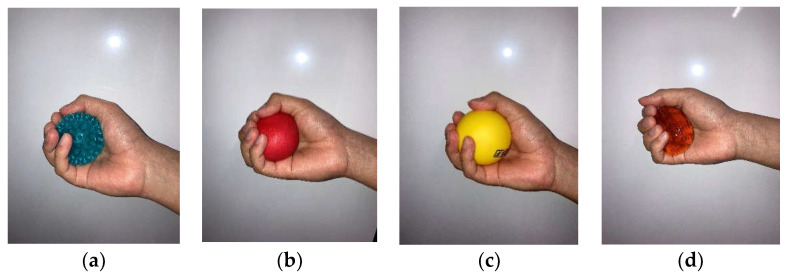
Pictures of the objects. (**a**) Object 1: A ball with wards; (**b**) Object 2: A squeezable ball; (**c**) Object 3: A regular hard ball; (**d**) Object 4: Slime.

**Figure 2 brainsci-12-00924-f002:**
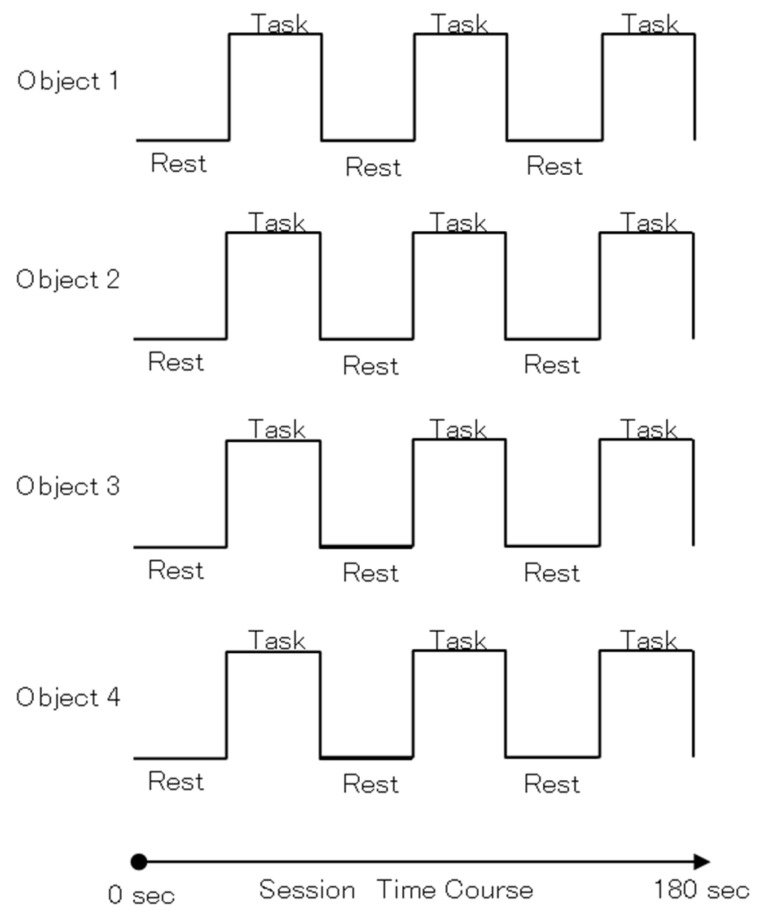
Block design of the stimulus task: 30 s, 30 s per task, three intervals per object.

**Figure 3 brainsci-12-00924-f003:**
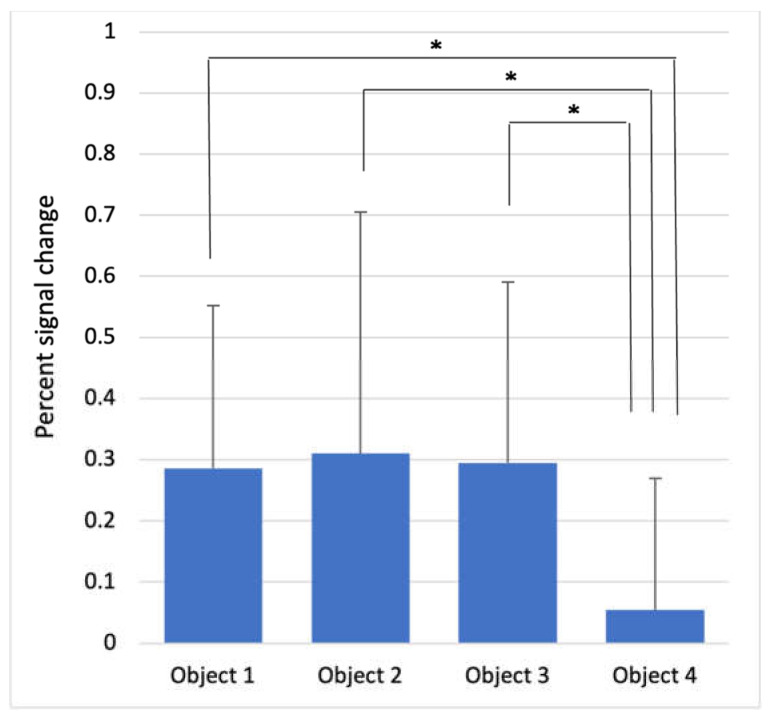
Percent signal change of each object in the anterior insula. * *p* < 0.05.

**Figure 4 brainsci-12-00924-f004:**
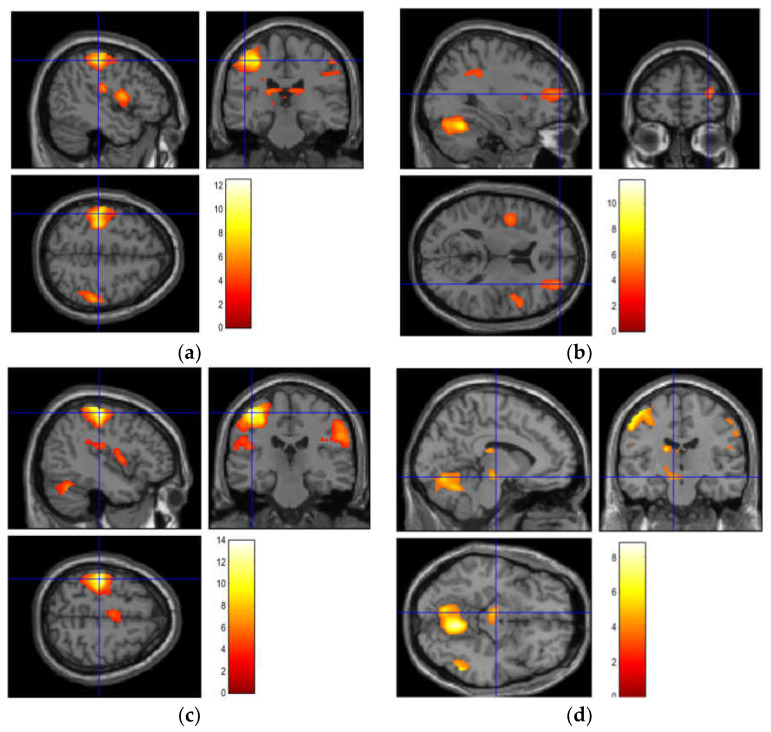
The images of the brain activation while grasping each object. (**a**) Object 1: Left postcentral gyrus (**b**) Object 2: Right middle frontal gyrus; (**c**) Object 3: Left postcentral gyrus; (**d**) Object 4: Left ventral diencephalon.

**Table 1 brainsci-12-00924-t001:** Results of the questionnaire for each object.

		Completely Disagree	Disagree	Undecided	Agree	Strongly Agree
Object 1	Comfort	17	3	2	3	0
	Hard	0	0	0	3	22
	Pain	0	0	0	5	20
	Easy to grasp	5	7	3	10	0
Object 2	Comfort	2	4	1	5	13
	Hard	17	5	0	2	1
	Pain	23	1	0	0	1
	Easy to grasp	0	1	1	8	15
Object 3	Comfort	6	9	6	3	1
	Hard	0	2	0	11	12
	Pain	19	6	0	0	0
	Easy to grasp	5	4	4	8	4
Object 4	Comfort	7	7	1	7	3
	Hard	25	0	0	0	0
	Pain	25	0	0	0	0
	Easy to grasp	18	6	1	0	0

**Table 2 brainsci-12-00924-t002:** Significantly activated areas and their T-values and coordinates when grasping each object.

	Cluster Size (voxels)	Cluster *p*-Value (FWE)	T-Value	Z-Score	x {mm}	y {mm}	z {mm}	Hemisphere	Locations
**Object 1**	3797	<0.001	12.43	6.96	22	−52	−26	Right	Lateral hemisphere of the cerebellum
	1989	<0.001	9.18	6.02	−46	−26	52	Left	Postcentral gyrus
	502	0.004	6.97	5.15	−20	−30	16	Left	Thalamus proper
	827	<0.001	6.62	4.98	−46	4	10	Left	Precentral gyrus
	539	0.003	6.39	4.88	52	−32	52	Right	Supramarginal gyrus
	394	0.012	5.59	4.46	26	40	16	Right	Middle frontal gyrus
	635	0.001	5.49	4.41	48	6	2	Right	Anterior insula
**Object 2**	5231	<0.001	11.77	6.79	20	−56	−22	Right	Lateral hemisphere of the cerebellum
	2557	<0.001	9.53	6.14	−36	−28	46	Left	Precentral gyrus
	2520	<0.001	7.18	5.24	56	−18	40	Right	Supramarginal gyrus
	715	<0.001	5.7	4.52	58	10	24	Right	Precentral gyrus
	337	0.019	5.14	4.2	32	54	12	Right	Middle frontal gyrus
**Object 3**	2642	<0.001	13.91	7.3	−42	−24	58	Left	Postcentral gyrus
	4589	<0.001	9.34	6.08	18	−54	−22	Right	Lateral hemisphere of the cerebellum
	2219	<0.001	7.91	5.55	60	−18	36	Right	Supramarginal gyrus
	397	0.014	5.13	4.2	−44	0	8	Left	Anterior insula
	378	0.017	5.11	4.19	−4	−10	54	Left	Supplementary motor cortex
**Object 4**	3729	<0.001	8.78	5.88	16	−54	−20	Right	Lateral hemisphere of the cerebellum
	2140	<0.001	7.5	5.38	−52	−20	48	Left	Postcentral gyrus
	395	0.004	7.4	5.34	−18	−20	18	Left	Thalamus proper
	281	0.019	6.44	4.9	56	−56	−10	Right	Inferior temporal gyrus
	1417	<0.001	6.16	4.76	54	−24	48	Right	Supramarginal gyrus
	222	0.044	5.69	4.52	−8	−16	−12	Left	Ventral diencephalon

**Table 3 brainsci-12-00924-t003:** Percent signal change in the brain areas according to each object.

	Object 1	Object 2	Object 3	Object 4	F-Value	*p*-Value
Precentral gyrus	0.358 ± 0.483	0.446 ± 0.424	0.234 ± 0.369	0.221 ± 0.380	1.716	0.169
Postcentral gyrus	1.124 ± 0.363	1.251 ± 0.782	1.128 ± 0.415	0.790 ± 0.593	2.640	0.054
Anterior insula	0.286 ± 0.266	0.310 ± 0.395	0.295 ± 0.296	0.054 ± 0.215	4.278	<0.001
Lateral hemisphere of the cerebellum	0.801 ± 0.401	0.968 ± 0.392	0.874 ± 0.527	0.645 ± 0.438	2.490	0.065
Ventral diencephalon	0.028 ± 0.252	0.106 ± 0.335	0.100 ± 0.289	0.206 ± 0.215	1.834	0.146

## Data Availability

The raw data supporting the conclusions of this article will be made available by the authors, without undue reservation.

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
