# Peer review of "Relationship between Tactile Sensation, Motor Activity, and Differential Brain Activity in Young Individuals"

_brainsci, 2022, doi:10.3390/brainsci12070924_

Round 1

Reviewer 1 Report

Please, see the attached pdf document.

Author Response

Dear Reviewer 1,

We would like to thank the reviewer for his/her constructive critique to improve the manuscript. We have made every effort to address the issues raised and to respond to all comments. Please find below the detailed, point-by-point responses to the reviewer's comments. We hope that our revisions would meet the reviewer’s expectations.

Introduction

(1)In particular, the conclusions of the Introduction (lines 64 - 68) seem to suggest a strategy based on pure data phishing, which is not acceptable at all. Namely, the authors “observed and analyzed the brain regions which showed activation” (line 69), without specifying any hypothesis about such activations. The authors should detail their hypotheses better as well as include a comprehensive overview of the study's objectives.

Response: In accordance with the reviewer’s comment, we have added the following part to the revised manuscript:

“The purpose of this study was to observe brain activity whilst the participants grasped four different shaped objects that were expected to cause different activations. We mainly focused on observing two different aspects at the same time: 1) brain activity during motor activity (grasping), and 2) brain activity produced by different sensations (comfort, hard, painful, and easy to grasp), which we expected to occur in the participants when they grasped different objects. We observed and analyzed the brain regions that showed activation. Moreover, through observing different types of brain activation, we aimed to verify the “action appraisal” mechanism, which is considered to be the act of integrating multiple different sensory information and connecting it to actual action [16]. Information generated from executing the task for the four objects (such as motion of moving the hand, sensation of touching the object, and emotion produced by gripping the object) should be collected and evaluated to verify such mechanism.” (Lines 64–75)

Materials and methods

(2)The authors should indicate all sociodemographic details of the sample, including participants’ age (mean and SD), gender, education, and handedness (and how it was calculated).

Response: In accordance with the reviewer’s comment, we have revised the sociodemographic details of the participants. The revised part is as follows:

“This study included 26 right-handed healthy, non-psychiatrically impaired individuals in their 20s (11 male and 15 female individuals; mean age, 21.0 years; standard deviation, 0.8 years) who had no experience with undergoing fMRI examination while grasping different types of objects.” (Lines 78–81)

 (3)In Figure 1, objects (a), (b) and (c) are grasped with the left hand but object (d) is grasped with the right hand. Were the participants checked for laterality and handedness?

Response: In accordance with the reviewer’s comment, we have revised the images presented in Figure 1.

(4)The authors should provide all the details about their sample. Also: has a statistical power study been preliminarily conducted? If not, the authors should report their convenience sample as a limitation of the study. In this case, I would suggest toning down the claims of generalization of their work, by including in the title or in the abstract an indication of "preliminary findings".

Response: In accordance with the reviewer’s comment, we have revised the title, abstract, and some parts of the Discussion section accordingly. In particular, we have emphasized that we included young participants in our study.

(5)How were the objects chosen? Did a jury evaluate them about what the authors indicate as "sensation/feeling"? The authors include a post-hoc questionnaire.

Response: In accordance with the reviewer’s comment, we have added the detailed substance material and size of each object as follows:

“First, we selected different types of objects that were expected to give the participants different feelings when they grasped them. We chose four different types of objects (Figure 1): Object 1, a ball with warts (Φ6cm, weight 40 g, PVC material); Object 2, a squeezable ball (Φ6 cm, weight 132 g, silicone); Object 3, a regular hard ball (Φ6 cm, weight 73 g, Rubber); and Object 4, Slime (Φ6 cm, weight 118 g). We also estimated the different kinds of feelings each object would produce: Object 1 (a ball with warts), comfort or pain; Object 2 (a squeezable ball), comfort; Object 3 (a regular hard ball), hard; and Object 4 (slime), discomfort. These were expected to show different activity in different brain regions.” (Lines 87–94)

(6)Were the objects preliminarily evaluated? It would be appropriate to specify these aspects in the Methods.

Response: As per the reviewer’s insightful suggestion, we have revised this part accordingly.

(7)The spatial arrangement of the objects in relation to the participants’ placement in the scanner is unclear. The authors state that the participants could not see the objects. However, the arrangement is not clear. An image would be helpful.

Response: In accordance with the reviewer’s comment, we have added a detailed explanation to the revised manuscript as follows:

“Before entering the MRI room, we instructed the participants to grasp the dynamometer at a grasping power of 10 kg and a pace of 100 beats per min (bpm), as set by the metronome, to rehearse the activity before the imaging experiment. After entering the MRI room, the participants were placed in the supine position on the MRI bed with their arms extended along the body side. The right palm was rotated outward to grip each of the four objects in turn, while the left hand remained in line with the body and was not used for the gripping tasks. Before each task, the experiment assistant entered the MRI room and placed the object on the participant’s right hand palm.” (Lines 110–118)

Results

(8)The MNI coordinates that are reported in Table 3 for Object 4 concerning the left SMG are wrong. Indeed, x = -8, y = -16, z = -12 do not correspond to the left SMG. Please, correct it.

Response: In accordance with the reviewer’s comment, we have revised the items and numbers of Object 4 in Table 3. We had made a mistake in the initial draft that the left SMG was strongly activated; however, it was the left ventral DC. We have corrected this mistake in the revised manuscript.

(9)How do the authors correct their results for multiple comparisons? Are the results FDR/FWE corrected? Please, specify the authors’ correction strategies in both methods and results section.

Response: In accordance with the reviewer’s comment, we have specified the correction strategies in “2.5 fMRI Data Analyses” subsection. The initial threshold for the voxel size was set to uncorrected p<0.001. Clusters were considered significant at p < 0.05, cluster-corrected for family wise error (FWE).

Discussion

(10)The authors extensively discuss their results concerning the activations in sensorimotor regions when participants manipulate objects. However, this is not a surprise nor a new finding in the literature. Predicting the activation of motor areas when participants grasp an object is quite intuitive. Surprisingly, the authors give less emphasis to the modulation of the frontoparietal network which they found when object characteristics change in terms of physical properties, namely what the authors summarized under the label of "sensation”.

In particular, it seems to me that the activations the authors found vary as a function of the object’s type, namely as an effect of the differences in the physical properties of the objects. The authors may therefore find useful the literature on technical reasoning, in the context of action semantics. Such literature highlights the role of a wide fronto-parieto-temporal network in action understanding and specifically the inferior parietal cortex in reasoning about physical properties. Most interestingly left and right SMG (namely BA40, a crucial region involved in physical understanding) seems to be active in Object 1 and Object 4, namely two completely different objects in terms of physical characteristics. I think this paper may represent an interesting step forward in the direction of the above-mentioned theoretical models. Concerning the integrations between different kinds of knowledge, namely semantic vs. motor vs. technical (physical understanding) contents, the authors may find useful the literature on the concept of “action reappraisal” (see below).

Please, note that the implicit evaluation of the physical components of objects involves most of the regions underlined by the authors. Therefore, I suggest the authors reduce the emphasis on the motor characterization of the effects they found, thus focusing on the relative contribution of their findings in such a specific brain network modulation / theoretical framework, namely technical reasoning and action reappraisal. In fact, the authors report data that are very well consistent with these theoretical perspectives.

Response: In accordance with the reviewer’s insightful comment, we have made major corrections in the Discussion section. First, we have removed most of the sentences that just insisted the results of the activation of motor areas as a result of grasping the objects. Instead, we have added relatively large text related to the “action reappraisal” concept, which was verified by the fact that the objects generated different types of information amongst the participants and integrated to recognize the objects. We have also added observation against the SMGs and added Table 4.

Reviewer 2 Report

This paper focuses on an interesting topic and the results are also interesting. However, the manuscript might be improved before publication and I have some concerns below.

The preprocessing and analyzing methods were not well described, particularly in line 127, “After the preprocessing, we employed a general linear model GLM to confirm brain activity changes associated with the four different grasping tasks, using a block design” – the descriptions are too simple and more details are needed for readers to repeat the analyses.

In the analyses, did the authors perform multiple comparison corrections across voxels? I didn’t see any descriptions.

How did the authors control image quality (e.g., did them exclude participants with poor image quality or excessive head motion)?

Did the authors try to control for sex/age effects, which are important in neuroscience studies, in all analyses?

What are the definitions of “strongly activated” and “uniquely activated”?

How can the authors differentiate the brain activation related to motor activity and the activation related to tactile stimuli (grasping objects) since they were detected simultaneously, if my understanding is correct? Are the conclusions only hypothetical?

Line 230, “Future research using alternative of improved methods would be needed” – In my opinion the analyses of dynamic functional connectivity (dFC) should be mentioned here. Investigating dFC is important because it can capture information which cannot be assessed by conventional static FC analysis methods. Altered dFC patterns within the sensorimotor areas have been also associated with multiple diseases (e.g., doi.org/10.1016/j.nicl.2020.102163, doi.org/10.3389/fpsyt.2020.00422).

Author Response

Dear Reviewer 2,

We would like to thank the reviewer for his/her constructive critique to improve the manuscript. We have made every effort to address the issues raised and to respond to all comments. Please find below the detailed, point-by-point responses to the reviewer's comments. We hope that our revisions would meet the reviewer’s expectations.

(1)The preprocessing and analyzing methods were not well described, particularly in line 127, “After the preprocessing, we employed a general linear model GLM to confirm brain activity changes associated with the four different grasping tasks, using a block design” – the descriptions are too simple and more details are needed for readers to repeat the analyses.

Response: In accordance with the reviewer’s comment, we have supplemented the preprocessing and analyzing methods in “2.5. fMRI Data Analyses” subsection. The revised part is as follows:

“The fMRI data were preprocessed and analyzed using Statistical Parametric Mapping 12 (Wellcome Trust Center for Neuroimaging) in Matlab (Mathworks Inc., Natick, MA, USA). Slice timing correction was used to correct the time difference and, then, realignment was used to correct the displacement caused by motion. In addition, a co-register was used to compare the structural images with the fMRI images. The co-register was corrected for misalignment between structural and functional images and the data were preprocessed by normalizing each participants’ brain to a template of the Montreal Neurological Institute coordinate system of a standard brain. Normalized images were smoothed using a Gaussian kernel of 8 mm. After preprocessing, we employed a general linear model GLM to confirm brain activity changes associated with the four different grasping tasks, using a block design. Contrast images were created at first level (single subject) for the following contrast: (1) Object 1 = 1, rest = 0; (2) Object 2 = 1, rest = 0; (3) Object 3 = 1, rest = 0 (4); Object 4 = 1, rest = 0. The head motion parameters obtained from the preprocessing step were included as regressors in each condition to minimize the effect of participant’s head motion artifacts. For group analysis (second level), a one-sample t-test was performed using the aforementioned four contrasts. The initial threshold for the voxel size was set to uncorrected p < 0.001. Clusters were considered significant at p < 0.05, cluster-corrected for family wise error. Each object was analyzed separately.” (Lines 138–155)

(2)In the analyses, did the authors perform multiple comparison corrections across voxels? I didn’t see any descriptions.

Response: In accordance with the reviewer’s comment, we have added descriptions for the multiple comparison corrections across voxels. The revised part is as follows:

“The initial threshold for the voxel size was set to uncorrected p < 0.001. Clusters were considered significant at p < 0.05, cluster-corrected for family wise error. Each object was analyzed separately.” (Lines 153–155)

(3)How did the authors control image quality (e.g., did them exclude participants with poor image quality or excessive head motion)

Response: In accordance with the reviewers comment, we have added the following sentence to the revised manuscript:

“The head motion parameters obtained from the preprocessing step were included as regressors in each condition to minimize the effect of participant’s head motion artifacts.” (Lines 150–152)

(4)Did the authors try to control for sex/age effects, which are important in neuroscience studies, in all analyses?

Response: In accordance with the reviewer’s comment, we have revised the sociodemographic details of the participants. The revised part is as follows:

“This study included 26 right-handed healthy, non-psychiatrically impaired individuals in their 20s (11 male and 15 female individuals; mean age, 21.0 years; standard deviation, 0.8 years) who had no experience with undergoing fMRI examination while grasping different types of objects.” (Lines 78–81)

(5)What are the definitions of “strongly activated” and “uniquely activated”?

Response: In accordance with the reviewers comment, we have added the following part to the revised manuscript :

“The coordinates of the areas that were (1) strongly activated, which was defined as the brain region that showed a reaction percentage of ≥50%, and (2) uniquely activated, which was defined as the brain region that showed activity only for a specific object within the four objects, are presented in Table 3.” (Lines 177–180)

(6)How can the authors differentiate the brain activation related to motor activity and the activation related to tactile stimuli (grasping objects) since they were detected simultaneously, if my understanding is correct? Are the conclusions only hypothetical?

Response: In accordance with the reviewers comment, we have added the following part to the revised manuscript:

“In this research, various pieces of information for the four objects, the motion of moving the hand, the sensation of touching the object, and the emotion produced by gripping the object, are considered to be integrated to recognize the objects. In particular, the supramarginal gyri (SMGs) (BA40) are activated in Objects 1 and 4, even if they are different objects (Table 4). SMGs comprise the IPL, which integrates somatosensory, visual, and auditory perception and is said to be involved in object identification and spatial perception. It is presumed that the SMG was activated by the multiple sensory information in this study.” (Lines 251–258)

(7)Line 230, “Future research using alternative of improved methods would be needed” – In my opinion the analyses of dynamic functional connectivity (dFC) should be mentioned here. Investigating dFC is important because it can capture information which cannot be assessed by conventional static FC analysis methods. Altered dFC patterns within the sensorimotor areas have been also associated with multiple diseases (e.g., doi.org/10.1016/j.nicl.2020.102163, doi.org/10.3389/fpsyt.2020.00422).

Response: In accordance with the reviewers comment, we have added the following part to insist the effectiveness of analyzing dFC:

“Future research using alternative of improved methods would be needed to further investigate the correlation between brain activity, motor activities, and sensation. First, investigating dynamic functional connectivity (dFC) within the results shall be effective and important, as such method can capture information that cannot be evaluated by conventional static FC analysis methods. Moreover, analyzing and investigating dFC patterns within the sensorimotor areas can also be associated with multiple types of cerebral neurosis.” (Lines 269–275)

Round 2

Reviewer 1 Report

The authors responded to all my comments in a satisfactory way. Therefore, I recommend the publication of the manuscript in this current version. Thanks for the opportunity to review this manuscript. 

Author Response

Please kindly see the attachment. (It is the same as the previous replies)

Reviewer 2 Report

None

Author Response

(The authors gave the same response as above.)
